# Intersections between Copper, β-Arrestin-1, Calcium, FBXW7, CD17, Insulin Resistance and Atherogenicity Mediate Depression and Anxiety Due to Type 2 Diabetes Mellitus: A Nomothetic Network Approach

**DOI:** 10.3390/jpm12010023

**Published:** 2022-01-01

**Authors:** Hussein Kadhem Al-Hakeim, Hadi Hasan Hadi, Ghoufran Akeel Jawad, Michael Maes

**Affiliations:** 1Department of Chemistry, College of Science, University of Kufa, Najaf 54001, Iraq; headm2010@yahoo.com (H.K.A.-H.); hhadi0615@gmail.com (H.H.H.); ggmmyazhra19378@gmail.com (G.A.J.); 2Department of Psychiatry, Medical University of Plovdiv, 4002 Plovdiv, Bulgaria; 3Department of Psychiatry, Faculty of Medicine, Chulalongkorn University, Bangkok 10330, Thailand; 4IMPACT Strategic Research Centre, School of Medicine, Deakin University, P.O. Box 281, Geelong, VIC 3220, Australia

**Keywords:** depression, mood disorders, inflammation, oxidative stress, nitrosative stress, neuro-immune, antioxidants, psychoneuroimmunology

## Abstract

Type 2 diabetes mellitus (T2DM) is frequently accompanied by affective disorders with a prevalence of comorbid depression of around 25%. Nevertheless, the biomarkers of affective symptoms including depression and anxiety due to T2DM are not well established. The present study delineated the effects of serum levels of copper, zinc, β-arrestin-1, FBXW7, lactosylceramide (LacCer), serotonin, calcium, magnesium on severity of depression and anxiety in 58 men with T2DM and 30 healthy male controls beyond the effects of insulin resistance (IR) and atherogenicity. Severity of affective symptoms was assessed using the Hamilton Depression and Anxiety rating scales. We found that 61.7% of the variance in affective symptoms was explained by the multivariate regression on copper, β-arrestin-1, calcium, and IR coupled with atherogenicity. Copper and LacCer (positive) and calcium and BXW7 (inverse) had significant specific indirect effects on affective symptoms, which were mediated by IR and atherogenicity. Copper, β-arrestin-1, and calcium were associated with affective symptoms above and beyond the effects of IR and atherogenicity. T2DM and affective symptoms share common pathways, namely increased atherogenicity, IR, copper, and β-arrestin-1, and lowered calcium, whereas copper, β-arrestin-1, calcium, LacCer, and FBXW7 may modulate depression and anxiety symptoms by affecting T2DM.

## 1. Introduction

Diabetes mellitus (DM) is a major public health issue with an increasing epidemic worldwide, accounting for 11.3 percent of all deaths [1]. In 2017, there were 451 million people aged over 18 years with DM worldwide, and it is with expected that this figure will increase to reach 693 million by the year 2045 [2]. Type 2 DM (T2DM) is initiated by insulin resistance (IR) in target tissues, high circulating insulin levels, β-cell dysfunction and subsequent β-cell failure [3]. IR is a condition whereby insulin-sensitive target tissues, such as adipose tissue, pancreas, skeletal muscles, and liver, do not react adequately to the physiological activities of insulin [4]. IR is part of the metabolic syndrome (MetS) cluster, which also involves aberrations in lipid profile, such as hypertriglyceridemia and low high density lipoprotein (HDL) cholesterol, abdominal obesity, and high blood pressure [5].

There is a significant comorbidity between mood disorders including major depressive and bipolar disorder and T2DM and MetS-associated features, including atherogenicity and IR [6,7,8,9,10,11]. According to the WHO (2017), T2DM is frequently accompanied by mood disorders and a systematic review and meta-analysis showed that the prevalence of depression in T2DM is around 25% [12]. For example, 10.6% of 2783 T2DM patients suffer from MDD and 17.0% from moderate to severe depressive symptoms [13]. Moreover, a meta-analysis reported that depression is associated with a significant elevated risk of T2DM and that the latter is associated with a slightly increased risk of depression [14], indicating that there are bidirectional relationships between T2DM and mood disorders. Nevertheless, mood disorders are more strongly associated with atherogenicity indices than with IR [10].

The comorbidity between mood disorders (either MDD or BD) and T2DM may be explained by multiple overlapping pathways, including IR, atherogenicity, activation of immune-inflammatory and nitro-oxidative stress pathways, an acute phase response, complement activation, T helper (Th)-17 activation, lowered antioxidant levels, mitochondrial dysfunction, and breakdown of the blood-brain-barrier (BBB) and the gut tight junctions barriers (leaky BBB and gut) [11]. Moreover, lowered plasma albumin, an inflammatory marker, predicts T2DM [15] and is a hallmark of MDD [16]. The same shared pathways underpin the comorbidity between mood disorders and MetS [17,18,19,20,21]. Shared biomarkers of mood disorders, T2DM, and MetS comprise increased levels of pro-inflammatory cytokines, malondialdehyde (MDA, indicating lipid peroxidation), nitric oxide (NO) metabolites (indicating increased NO production), and advanced protein oxidation products (AOPPs), and lowered paraoxonase (PON)-1 activity [10,11,22,23]. Nevertheless, the biomarkers of affective symptoms (depression and anxiety) in T2DM are not well established and, consequently, research should focus on the up- or downstream biomarkers of the above-mentioned pathways that play a role in depression and anxiety due to T2DM. Therefore, the present research focuses on the role of β-arrestin-1, FBXW7, CD17, copper, zinc, calcium, and magnesium in depression/anxiety due to T2DM.

β-arrestin-1 controls β-cell functions and survival and mediates insulin secretion [24]. β-arrestin-1 is reduced in the white blood cells of depressed patients while antidepressants increase β-arrestin-1 levels in leukocytes of depressed patients and rat brain [25,26]. β-arrestin-1 desensitizes G-protein coupled receptor (GPCR) signaling, which plays a role in depression [26] and links GPCRs to downstream pathways including ERK1/2 [25], which plays a role in T2DM and mood disorders [27,28]. Moreover, β-arrestin-1 interacts with cAMP-Response Element Binding protein (CREB), thereby regulating neurobiological processes including cell replication, survival, and plasticity [29]. Aberrations in CREB-mediated transcription are associated with depression, anxiety, and cognitive impairments [30].

FBXW7 (F-box/WD repeat-containing protein 7), also known as human CDC4, is an E3 ubiquitin ligase which targets cyclin E for ubiquitin-mediated degradation [31]. Lowered FBXW7 in animal models and humans is associated with hyperglycemia, IR, and the development of T2DM [32]. In primary neurons, peroxisome proliferator-activated receptor gamma coactivator 1-alpha (*PGC**-**1α*) and mechanistic target of rapamycin complex 1 (mTORC1) are FBXW7 substrates and regulate glucose homeostasis [33,34]. Moreover, FBXW7 regulates neurogenesis by antagonizing c-Jun and Notch, which is a key regulator of neuronal differentiation and synaptic plasticity [35]. Furthermore, lowered FBXW7 is accompanied by aberrations in stem cell differentiation in the brain [36]. FBXW7 also regulates the turnover and stability of disrupted in schizophrenia (DISC) which orchestrates neural cell signaling and differentiation [37]. Aberrations in neurogenesis, mTOR, DISC, PGC-1α are thought to play a role in neuroprogressive mental disorders, including mood disorders [36,38,39].

Lactosylceramides (LacCer or CD17) form lipid rafts on the membrane of neutrophils and are involved in chemotaxis, phagocytosis, and superoxide generation [40,41]. LacCer activates NADPH oxidase, which produces superoxide radicals (O_2_^-^) [42] and inducible nitric oxide synthase (iNOS) and NO [43]. Mild diabetes is accompanied by an increased conversion of glucosylceramide (GluCer) to LacCer [44]. Ceramides including LacCer are significantly elevated in depressed patients [45,46,47] and the activity of sphingomyelinase may be increased in mood disorders [45].

Alterations in peripheral serotonin levels may be another shared biomarker among T2DM and affective symptoms. Serotonin is synthesized within β-cells [48] and is stored together with insulin in their secretory β-granules [49], and it is co-released when pancreatic islets are stimulated by glucose [50]. Lowered levels of serotonin including in platelets are observed in depressed patients [51]. Other biomarkers that may link mood disorders and T2DM comprise copper, zinc, calcium, and magnesium. Recent reviews and meta-analyses reported that zinc and copper are involved in the pathogenesis of diabetes and IR [52] while reduced zinc and increased copper levels are hallmarks of depression [53,54]. A meta-analysis showed that increased plasma calcium levels are associated with T2DM [55], whereas Al-Dujaili et al. [56] established lowered calcium levels in major depression. Magnesium deficiency and/or low magnesium dietary intake may cause IR, glucose tolerance, T2DM and MetS [55,57] and depression [58]. In patients with atherosclerosis and unstable angina, IR is associated with lowered magnesium and zinc and increased copper, whereas comorbid depression is associated with IR coupled with lowered zinc and increased calcium [59]. Nevertheless, no research has delineated the effects of the above-mentioned biomarkers on the severity of affective symptoms including depression and anxiety due to T2DM.

Hence, the present study was conducted to delineate the effects of serum levels of β-arrestin-1, FBXW7, CD17, serotonin, albumin, calcium, magnesium, zinc, and copper on severity of depression and anxiety above and beyond the effects of IR and atherogenicity in T2DM.

## 2. Subjects and Methods

### 2.1. Subjects

The current case-control study recruited 58 T2DM male patients and 30 age, BMI and education matched healthy controls. We selected male subjects to exclude possible effects of the female hormonal status and the menstrual cycle. The subjects were recruited at Al-Sader medical city, Najaf governorate, Iraq during the period November 2020 till January 2021. The diagnosis of T2DM was made using the World Health Organization criteria [60,61] and fasting plasma glucose (FPG) ≥ 7.0 mM and glycated hemoglobin (HbA1c) > 6.5%. Included were T2DM patients with and without affective symptoms. Nevertheless, we excluded any subjects with psychiatric axis-I diagnosis according to DSM-5 criteria except T2DM patients with a “mood disorder due to a general medical condition, depressed mood or diminished interest or pleasure in all or almost all activities”. Moreover, patients were excluded if their serum FPG was greater than 25 mmol/L and their fasting insulin was greater than 400 pM, to comply with the requirements of the HOMA calculator program. In addition, we excluded subjects who had overt diabetic comorbidities such as cardiac failure, liver disease, or kidney disease. We also excluded patients (a) who are receiving metformin because the latter may affect IR [62] and insulin sensitivity [63]; and (b) with an albumin/creatinine ratio > 30 mg/g [64]. All participants had serum CRP concentrations <6 mg/dL excluding people with overt inflammation. Before taking part in the study, all participants gave written informed consent. Approval for the study was obtained from the IRB of the University of Kufa (T1375/2020), which complies with the International Guidelines for Human Research protection as required by the Declaration of Helsinki.

### 2.2. Assessments

Severity of depression was assessed using the total score on the 17-item Hamilton Depression Rating Scale (HDRS) score [65] and severity of anxiety was assessed using the total score on the Hamilton Anxiety Rating Scale (HAM-A) [66]. As explained previously [67] we computed three HDRS subdomain scores, i.e., (a) key depressive symptoms (key_HDRS) computed as the sum of depressed mood + feelings of guilt + suicidal ideation + loss of work and activities; (b) physiosomatic symptoms (physiosomatic_HDRS) computed as the sum of anxiety somatic + somatic symptoms, gastrointestinal + somatic symptoms, general + genital symptoms + hypochondriasis; and c) melancholic symptoms (melancholia_HDRS) computed as sum of insomnia late + psychomotor retardation + diurnal variation + loss of weight. As reported by Almulla et al. (2021) [67], we computed two different HAM-A subscores, i.e., key anxiety symptoms (key_HAM-A) as sum of anxious mood + tension + fears + anxious behavior at interview; and HAM-A physiosomatic symptoms (physiosomatic_HAM-A) computed as sum of somatic muscular + somatic sensory + cardiovascular symptoms + respiratory symptoms + gastrointestinal symptoms + genitourinary symptoms + autonomic symptoms. Body mass index (BMI) was determined by dividing weight in kilograms by height in meters squared.

After an overnight fast, five milliliters of blood were drawn in the morning from patients and controls. After full clotting, blood was centrifuged at 3000 rpm for 10 min to separate serum, which was then frozen at −80 °C until thawed for assay. Serum copper and zinc were measured spectrophotometrically using kits supplied by Spectrum Diagnostics Co. (Cairo, Egypt). LacCer (CD17), serotonin, and FBXW7 were measured in sera by *ELISA kits* supplied by *Melsin* Medical Co, Jilin, China, while β-arrestin-1 was estimated using ELISA kits provided by Bioassay Technology Laboratory (Shanghai, China). The processes were carried out precisely as prescribed by the manufacturer, with no deviations. Serum insulin was measured by commercial ELISA sandwich kit supplied by DRG^®^ International Inc., New Jersey, USA. The sensitivities of the kits were 12.22 pM for insulin, 0.05 ng/mL for β-arrestin-1, 1.0 ng/mL for serotonin, 0.1 ng/mL for LacCer, and 0.1 ng/mL for FBXW7. Sera were diluted 1:4 to estimate LacCer levels. Fasting serum levels of albumin, calcium, magnesium, glucose (FBG), total cholesterol (TC), and triglycerides (TG) were measured spectrophotometrically by a ready for use kits supplied Spinreact^®^, Girona, Spain. Serum high-density lipoprotein cholesterol (HDLc) was determined after precipitating other lipoproteins with a reagent comprising sodium phosphotungstate and MgCl_2_. Then, the cholesterol contents in the supernatant were assessed using same kit of total cholesterol. Low-density lipoprotein cholesterol (LDLc) was calculated from Friedewald’s formula: LDLc = TC − HDLc–TG/2.19. The intra-assay coefficient of variation (CV) (precision within an assay) was < 10.0% for all ELISA assays. Serum CRP was measured using a kit supplied Spinreact^®^, Girona, Spain. The test is based on the principle of latex agglutination.

In the current study, two atherogenic indices were computed, namely z score of total cholesterol—z HDL cholesterol (zCastelli), which reflects the Castelli risk index 1, and z triglyceride—z HDL cholesterol (zAIP), which reflects the atherogenic index of plasma (AIP) [68,69]. In the current study, there were significant correlation between zTC—z HDL and the Castelli risk index 1 (r = 0.995, *p* < 0.001, *n* = 88) and between zTG-zHDL and the AIP index (r = 0.904, *p* < 0.001, *n* = 98). In the current study, we also computed z unit-weighted composite scores reflecting IR as z glucose + z insulin (IRI), and β cell function as z insulin—z glucose (zβCell). We found significant correlations between IRI and HOMA2IR as defined with the HOMA2 Calculator^©^ (Diabetes Trials Unit, University of Oxford) (r = 0.881, *p* < 0.001, *n* = 88) and between zβCell and HOMA2B (r = 0.781, *p* < 0.001, *n* = 88).

### 2.3. Statistical Analysis

We employed analysis of variance (ANOVA) to check differences in scale variables between sample groups. Analysis of contingency tables (chi-square test) was employed to assess associations among nominal variables. Correlation matrices based on Pearson’s product-moment were used to examine associations between biomarkers. We used automatic multiple regression analysis to define the significant biomarkers (β-arrestin-1, FBXW7, CD17, serotonin, albumin, calcium, magnesium, zinc, and copper) predicting the HDRS and HAM-A total and subdomain scores and checked whether these biomarkers had significant effects above and beyond the effects of the Castelli, AIP, IRI, and z β-cell indices (or FBG, insulin or the fatty acids) while allowing for the effects of age, education, and BMI. We employed an automatic stepwise (step-up) method with a p-to-enter of 0.05 and p-to-remove 0.06 while checking R^2^ changes, multivariate normality (Cook’s distance and leverage), multicollinearity (using tolerance and VIF), and homoscedasticity (using White and modified Breusch–Pagan tests for homoscedasticity). In case of heteroscedasticity, we used HC3 parameter estimates with robust standard errors. We also conducted automatic binary logistic regression analysis with T2DM as dependent variable and biomarkers as explanatory variables and calculated Odds ratios with 95% confidence intervals. For classification purposes, we used the random oversampling method to adjust the class distribution of the normal control class. All results were bootstrapped (5000 bootstrap samples) and the latter results are shown if the results are not concordant. All tests are two-tailed, with a *p* value of 0.05 used to determine statistical significance. Statistical analyses were carried out using IBM SPSS Windows version 25, 2017, manufactured by IBM Corp., New York, NY, USA.

Partial Least Squares (PLS) path analysis (SmartPLS) [70] was used to measure the causal association between biomarkers, T2DM (atherogenicity and IR) and the phenome of affective symptoms. The variables were entered as latent vectors (LVs) extracted from their reflective manifestations or as single indicators. We performed complete PLS path analysis on 5000 bootstrap samples only when the inner and outer models complied with quality data, namely (a) the overall model fit is accurate as indicated by SRMR < 0.08, (b) all LVs have an accurate construct validity as indicated by average variance extracted >0.5; composite reliability (>0.7), Cronbach’s alpha (>0.7), and rho_A (>0.8), (c) all outer model LV loadings are >0.666 at *p* < 0.001, (d) Monotrait–Heterotrait analysis indicates adequate discriminatory validity, (e) blindfolding shows that the construct cross-validated redundancies or communalities are adequate, and (f) Confirmatory Tetrad analysis indicates that the LV models are not mis-specified as reflective models. PLS predict with 10-fold cross-validation was employed to check the predictive performance when analyzing new data [71]. Prediction-Oriented Segmentation analysis, Multi-Group Analysis and Measurement Invariance Assessment were employed to examine compositional invariance.

A priori estimation of the required study sample showed that at least 70 individuals should be included to obtain a power of 0.8 with an effect size of 0.2, alpha level of 0.05, and 5 predictors in a linear multiple regression model. The same power analysis can be applied when examining PLS path analysis indicating that the power of this PLS analysis was >0.8 [72].

## 3. Results

### 3.1. Socio-Demographic and Clinical Characteristics

Table 1 shows the socio-demographic and clinical data of subjects with a normal IRI, a medium IRI, and a high IRI. There were no significant differences in age, BMI, education, residency, marital status, and employment among these three groups. The family history of DM was significantly different among those three groups with an increasing frequency from the normal IRI -> medium IRI -> high-IRI group. In the patient group, 12 patients were drug free, 19 were on diabetes diet, and 27 were treated with glibenclamide 5 mg/day. The treatment regimen showed a significant difference among groups with the highest ratio of subjects treated with glibenclamide in the high IRI group as compared with the two other groups. Nevertheless, the drug/diet state of the patients was entered in all multiple linear regression analyses as explanatory dummy variables. However, neither diabetes diet nor glibencamide showed a significant effect on the dependent variables. FBG, insulin, IRI, and zAIP were significantly different between the three study groups and their values increased from normal IRI -> medium IRI -> high IRI. The zβcell index, and HDL- and LDL cholesterol showed no significant differences among the three study groups. Total cholesterol and TG were significantly higher in the medium and high IRI groups than in the normal IRI group. The zCastelli index was significantly higher in subjects with a high IRI as compared with those with a normal IRI.

### 3.2. Rating Scale Scores and Biomarkers in the IRI Subgroups

The total and subdomain scores on the HDRS and HAM-A and biomarker levels are presented in Table 2. The total HDRS and key HDRS scores were significantly higher in the high IRI group as compared with the two other IRI groups. The physiosomatic and melancholia subdomains of the HDRS were significantly higher in subjects with high IRI as compared with those with a normal IRI. The total HAM-A, key_HAM-A, and physiosomatic_HAM-A scores were increased in the high IRI group as compared with the medium and normal-IRI groups.

β-arrestin-1, serotonin, albumin, magnesium, calcium, copper, and zinc were not significantly different between the three IRI subgroups. FBXW7 was significantly decreased in the high IRI subgroup as compared with the two other groups, whereas sCD17 was significantly increased in the high IRI group as compared with the two other groups.

### 3.3. Prediction of HDRS Score Using Biomarkers

Table 3 shows the results of the two types of multiple regression analysis with the total and subdomain scores of the HDRS and HAM-A as dependent variables and the biomarkers listed in Table 2 (a type) and zCastelli, zAIP and IRI indices (b type) as explanatory variables while allowing for the effects of age, education, and BMI. Regression #1a shows that 47.8% of the variance in the total HDRS-17 score could be explained by CD17, copper and β-arrestin-1 (all positively associated) and calcium (inversely associated). Figure 1 shows the partial regression plot of the total HDRS score on β-arrestin-1 after adjusting for the variables listed in Table 3, regression #1a. In Regression #1b, 49.9% of the variance in the total HDRS-17 score could be explained by zCastelli index, copper, and β-arrestin-1 (all positively associated) and calcium (inversely associated). We found that 34.4% of the variance in the key_HDRS score could be explained by the regression on β-arrestin-1 and copper and negatively with calcium (Regression #2a). Regression #2b showed that 43.2% of the variance in the key_HDRS score could be explained by the regression on the zCastelli index, copper and β-arrestin-1 (all positively associated) and calcium (inversely associated). Regression #3a showed that 27.2% of Physiomatic_HDRS could be explained by the regression on FBXW7 and calcium (both inversely) and copper (positively associated). Figure 2 shows the partial regression plot of the physiosomatic_HDRS scores on calcium after adjusting for the other variables. Regression #3b shows that 28.2% of the variance in the physiomatic_HDRS could be explained by the regression on the IRI and copper (positively associated), and calcium (both inversely). Regression #4a shows that 26.5% of the variance in the melancholia_HDRS score could be explained by the regression on the copper and β-arrestin-1 (both positively associated) and zinc (negatively). Regression #4b shows that 35.8% of the variance in the melancholia_HDRS score could be explained by the regression on the zCastelli index, copper, albumin, and β-arrestin-1.

### 3.4. Prediction of Hamilton Anxiety Rating Scale (HAM-A) Scores

Table 4 presents the results of multiple regression analysis with the HAM-A total and subdomain scores as dependent variables and biomarkers as explanatory variables. Regression #1a shows that 47.7% of the variance in the HAM-A total score could be explained by the regression on copper, β-arrestin-1, and LacCer (all positively associated) and calcium (negatively). Figure 3 shows the partial regression plot of the total HAM-A score on serum copper after adjusting for the other variables in this regression. We found that 52.9% of the variance in HAM-A total score could be explained by the regression on β-arrestin-1, zCastelli, and copper (all positively associated) and calcium (negatively) (Regression #1b). Regression #2a shows that 46.6% of the variance in the key_HAM-A score could be explained by the regression on copper and β-arrestin-1 (all positively associated) and FBXW7, calcium and age (all negatively). We found that 49.4% of the variance in the key_HAM-A score could be explained by β-arrestin-1, zCastelli, and copper (all positively associated) and age, calcium, and FBXW7 (negatively) (Regression #4). Regression #3a show that 43.7% of the variance in the physiosomatic_HAM-A score could be explained by the regression on copper, LacCer, and β-arrestin-1 (positively associated) and calcium (negatively associated). We found that 47.4% of the variance in the physiosomatic_HAM-A score could be explained by the regression on copper, β-arrestin-1, and zAIP (all positively associated) and calcium (negatively) (regression #3b).

### 3.5. Prediction of T2DM Using Biomarkers

Table 5 shows the results of binary logistic regression analysis with T2DM as dependent variable and controls as the reference group. Binary logistic regression is used to predict the odds of being a patient using the biomarkers as independent variables. T2DM was significantly predicted by LacCer, copper, and serotonin (positively associated) and FBXW7 and calcium (inversely associated) with a Nagelkerke pseudo-R^2^ value of 0.803 and accuracy of 92.3% (sensitivity = 93.3% and specificity = 91.2%). Binary regression analysis also showed significant positive associations between T2DM and β-arestin-1 and an inverse association with magnesium (all after FDR p correction).

### 3.6. Biomarker Predictors of IRI, β-Cell Function, Castelli, and AIP Indices

Table 6 shows the outcome of different automatic multiple regression analysis with IRI, zβcell function, zCastelli or zAIP as dependent variables. Regression #1 shows that 14.6% of the variance in IRI was explained by the regression on FBXW7 and LacCer. Regression #2 shows that β-arrestin-1, calcium, copper, albumin, and a DM family history explained 45.0% of the variance in the zβcell index. In regression #3 we found that calcium, FBXW7, and a positive DM family history explained 29.5% of the variance in the zCastelli index. Regression #4 shows that 24.2% of the variance in the zAIP index could be explained by calcium, magnesium, and serotonin.

### 3.7. Results of PLS Path and PLS Predict Analysis

Figure 4 shows the final PLS model obtained after feature selection, prediction-oriented segmentation with multi-group analysis and PLSpredict analysis. T2DM was entered as a reflective LV extracted from FBG, T2DM, triglycerides, and total cholesterol (the mediator) and the depression and anxiety phenome was entered as a reflective LV extracted from key_HDRS, physiosomatic_HDRS, melancholia_HDRS, key_HAM-A, and physiosomatic_HAM-A (the endogenous variable). The other variables were entered as single indicators. The overall fit of the PLS model was adequate with SRMR = 0.051 and also the construct reliabilities of the LV were adequate, namely for the T2DM LV we found a composite reliability of 0.883, Cronbach α: 0.832, rho A: 0.934, and AVE: 0.655; and for the affective symptom LV a composite reliability: 0.915, Cronbach α 0.882, rho A 0.890, and AVE: 0.683. All outer model loadings on both LVs were > 0.708 at *p* < 0.0001. The construct cross-validated redundancies were adequate, namely the T2DM LLV: 0.245 and affective symptom LV: 0.410. Complete PLS path analysis performed on 5.000 bootstrap samples showed that 61.7% of the variance in the affective symptom LV could be explained by the regression on the T2DM LV, copper, β-arrestin-1, and calcium, while 44.4% of the variance in the T2DM LV was explained by copper, β-arrestin-1, LacCer (all positively) and calcium and FBXW7 (both negatively). There were specific indirect effects of copper (t = −2.72, *p* = 0.007), calcium (t = 2.54, *p* = 0.011), FBXW7 (t = 2.05, *p* = 0.040), and LacCer (t = 2.23, *p* = 0.026) on affective symptoms which were mediated by the T2DM LV. There were significant total effects (in descending order of importance) of copper (t = −4.39, *p* < 0.001), calcium (t = −4.19, <0.001), β-arrestin-1 (t = 3.35 *p* = 0.001), LacCer (t = 2.29, *p* = 0.026), age (t = 2.27, *p* = 0.023), and FBXW7 (t = −2.05, *p* = 0.040) on the affective symptom LV. PLS predict with 10-fold cross-validation [71] showed that a) the Q2 Predict values of the affective and T2DM LVs were positive, indicating that the PLS prediction error is smaller than that of the most naïve benchmark; (b) in comparison with the linear regression model, the PLS results showed a lower prediction error; and (c) all indicators except one have a lower RMSE compared to the naive linear regression benchmark indicating medium to strong predictive power. Prediction-Oriented Segmentation analysis followed by Multi-Group Analysis and Measurement Invariance Assessment showed that full compositional invariance was established.

## 4. Discussion

### 4.1. Biomarkers of T2DM

The first major finding of the current study is that copper, LacCer, (positively associated) and FBXW7 and calcium (inversely associated) predict around 44.4% of the variance in a composite score comprising IR and atherogenicity. Both copper and LacCer were the best predictors of this composite score and T2DM, although also magnesium contributed to the prediction of T2DM.

A recent meta-analysis reported increased copper levels in 1079 DM patients including those with T2DM and T1DM as compared with 561 healthy controls [73]. Tanaka et al. [74] showed that in diabetic C57BL/KsJ-db/db mice, serum copper was significantly increased in association with increased reactive oxygen species, while copper chelating attenuated IR and triglyceride levels and improved glucose intolerance. The increased copper levels in T2DM are positively associated with increased production of reactive oxygen species [75] and glycated hemoglobin A1C [76] and negatively with ceruloplasmin and thiol levels [77]. These findings indicate that copper-induced oxidative stress plays a key role in T2DM and glycemic control [78]. Although abnormal zinc metabolism may play a role in the onset and maintenance of diabetes and IR [52,79] we could not find significant associations between T2DM and lowered zinc.

In our study, LacCer appears to be a highly significant predictor of T2DM and IR as well. As reviewed in the Introduction, mild diabetes is accompanied by an increased conversion of GluCer to LacCer [44]. Our findings may be explained by the effects of LacCer on pattern recognition receptors leading to activated immune, oxidative and nitrosative pathways [40,41,42,43]. Moreover, LacCer is a direct cytosolic phospholipase alpha 2 activator (cPLA2α) by stimulating phosphorylation signals and attachment of the enzyme to substrate membranes [80], a phenomenon that is associated with activated inflammatory pathways. Furthermore, increased levels of serum LacCer predict cardiovascular disease progression and mortality above and beyond the effects of established risk factors [81].

We found that T2DM is also accompanied by lowered serum FBXW7, a factor protecting against IR and atherogenicity. For example, in ob/ob mice, FBXW7 expression decreases blood glucose and insulin levels, IR and glucose intolerance, and prevents expression of lipogenic genes and triglyceride accumulation [82]. Liver-specific FBXW7 knockout mice develop hyperglycemia, glucose sensitivity, and IR [82]. Moreover, FBXW7 controls fetuin-A expression in obese humans, where the increase in fetuin-A may lead to IR and T2DM [83,84]. Inactivation of FBXW7 causes the sterol regulatory element-binding protein 1SREBP1 to accumulate, and the expression of SREBP1 increases the expression of genes that are involved in lipid metabolism and synthesis of triglycerides [85]. On the other hand, hyperglycemia may suppress FBXW7 expression in renal mesangial cells, resulting in increments in inflammatory responses [86], and treatment of human kidney proximal tubular cells with glucose significantly lowers FBXW7 expression [87].

In our study, increased serum β-arrestin-1 is another biomarker of T2DM and lowered beta-cell function. Increased expression of wild-type β-arrestin-1 decreases insulin-induced degradation of insulin receptor substrate 1 (IRS-1), leading to increased insulin signaling, and attenuation of β-arrestin-1 enhances IRS-1 degradation, thereby accentuating cellular IR [88]. β-arrestin-1 may desensitize the glucagon-like peptide-1 (GLP-1) receptor on pancreatic cells, which potentiates glucose-stimulated insulin secretion [89]. Both β-arrestin-1 and -2 are critical regulators of immune-inflammatory responses and exert multiple effects on immune pathways, including macrophage, neutrophil and T lymphocyte, and Toll-Like Receptor functions, nuclear factor-κB activity, explaining that these molecules play a key role in many immune disorders, including atherosclerosis [90,91]. Overall, β-arrestin-1 shows anti-inflammatory effects, although this intracellular scaffolding protein also shows some pro-inflammatory effects in some models [92]. Furthermore, β-arrestin-1 reduces oxidative stress via Nrf-2 activation [93], although overexpression of β-arrestin in cardiac fibroblasts significantly elevates Nox-4 mitochondrial superoxide production in an ERK-dependent manner [94]. β-arrestin-1 shows a strong plasma membrane representation, and interactome analysis show that β-arrestin-1 interactome activity modulates many downstream signalers that play a role in ageing, including cell cycle regulation, G-protein associated functions, and opioid signaling [95]. By inference, in T2DM, β-arrestin-1 may play a role in the global reduction in homeostatic stability and/or compensatory mechanisms characterized by a switching of the metabolism to less efficient processes [95].

Magnesium deficiency is a risk factor for IR, diabetes, hypertension, atherogenicity, and cardiovascular disease [96] and has a detrimental effect on blood glucose homeostasis [97,98,99]. Our calcium findings do not concur with those of a meta-analysis showing that increased plasma calcium levels are associated with T2DM [55]. Nevertheless, in non-diabetic adults, calcium supplementation may reduce plasma glucose levels and IR [100].

In our study, increased levels of peripheral serotonin are associated with T2DM after considering the impact of copper, calcium, LacCer, and FBXW7. Previously, it was shown that inhibiting peripheral serotonin synthesis and signaling may effectively treat T2DM, obesity, and nonalcoholic fatty liver disease [101]. Moreover, genome-wide association studies found associations between genetic polymorphisms in tryptophan hydroxylase and serotonin receptors and obesity [102]. As described in the Introduction, serotonin may control glucose metabolism [103] and affect insulin secretion through serotonin receptors [104,105]. Intracellular serotonin, through serotonylating GTPases including Rab3a and Rab27a, regulates insulin secretion, resulting in serotonin and insulin co-release [106]. 

### 4.2. Biomarkers of Affective Symptoms Due to T2DM

The second major finding of this study is that we were able to construct a reliable and replicable nomothetic model of affective symptoms due to T2DM [72,107,108]). Our model provided a 10-fold cross-validated prediction in a holdout sample, indicating accurate predictive performance when analyzing new data. Direct predictors of affective symptoms were copper, calcium, and β-arrestin, whereas FBXW7 and LacCer showed indirect effects that were mediated by T2DM. Nevertheless, lowered zinc was a significant predictor of depressive melancholia scores.

Importantly, we found that increased atherogenicity had a greater impact on affective symptoms than IR. In this respect, primary mood disorders were consistently characterized by increased atherogenicity, whereas IR was not always associated with mood disorders [109,110]. There is now also evidence that atherogenicity, as indicated by an increased Castelli risk index 1 and AIP, significantly contributes to the pathophysiology of mood disorders via immune-inflammatory and nitro-oxidative pathways [111,112,113].

In our nomothetic model, the most important predictors of depression and anxiety symptoms were increased copper and lowered calcium. A recent meta-analysis showed increased copper levels in 1787 depressive patients versus 943 controls [54]. Copper toxicity may also cause anxiety, cognitive impairments, sleep disorders, and physiosomatic symptoms, including muscle pain and tension, autonomous symptoms, heart palpitation, headache, excessive perspiration, etc. [114]. The depressogenic and anxiogenic effects of serum copper may be explained by increased oxidative stress and lowering levels of serum zinc [115]. Nevertheless, there are also more recent studies showing that copper may be decreased or unchanged in patients with depression [115,116]. In our study, zinc was inversely associated with melancholia symptoms. Recent meta-analyses showed significantly lower zinc levels in depressed patients than in controls [117,118]. Lowered serum zinc predisposes towards immune-inflammatory and nitro-oxidative toxicity and, therefore, affective disorders [119].

The results of the present study are also in agreement with previous findings that lowered serum calcium is associated with major depression and with self-rated depression, irritability, agitation, and physiosomatic symptoms, including neuromuscular excitability [56,120]. Nevertheless, comorbid depressive ratings in unstable angina were not only associated with atherogenicity, IR, lowered zinc but also with increased IL-6 and calcium levels [121]. The latter may, however, be explained by specific effects of calcium mineralization of arteries and calcification of plaques in unstable angina [121].

Our results showing that β-arrestin-1 is positively associated with depressive symptoms contrasts, at first sight, with those of previous reports that β-arrestin-1 is reduced in the white blood cells of depressed patients and that reductions in β-arrestin-1 are significantly associated with the severity of depressive symptoms [25,122]. Those previous reports suggest that β-arrestin-1 is involved in the pathophysiology of affective disorders and in the pathways that mediate antidepressant effects [25,123,124]. Nevertheless, increased plasma β-arrestin levels may be ascribed to leakage of β-arrestin out of the tissues [125]. Therefore, our results suggest that increased β-arrestin-1 levels may be associated with the onset of affective symptoms due to T2DM following desensitization of GPCR signaling and GPCR-associated effects on ERK1/2 and CREB (see Introduction). Furthermore, β-arrestin-1 exerts stimulatory effects on microglia-mediated inflammation and STAT1 and nuclear factor-κB activation [126], processes that are associated with affective symptoms [127].

As described in our Introduction, LacCer and serum levels of other ceramides are significantly increased in mood disorders [45,46,47], which may in part be explained by increased sphingomyelinase activity [45]. Interestingly, ceramide and sphingomyelin species are involved in the pathophysiology of depressive symptoms in coronary artery disease [128]. Increased levels of pro-inflammatory cytokines and oxidized LDL (which play a role in depression, T2DM, and MetS) may increase the biosynthesis of LacCer, which consequently may induce oxygen-specific pathways, thereby generating reactive oxygen and nitrogen species as well as peroxynitrite, and inducing immune-inflammatory pathways [129,130]. Moreover, high ceramide levels in the hippocampus are associated with decreased neurogenesis, neuronal maturation, and neuronal survival [131] and increased amygdala sphingolipids are associated with anxiety-like behaviors in animal models [132].

This is a first report that lowered FBXW7 may be associated with depressive and anxiety symptoms, although these effects may be mediated via increased IR because lowered FBXW7 is associated with hyperglycemia, IR, and the development of T2DM [32]. Moreover, as explained in the Introduction, FBXW7 modulates some pathways that play a role in mood disorders, including neurogenesis, mTOR, DISC, and PGC-1α [36,38,39].

### 4.3. Limitations

The results of the current study should be interpreted regarding its limitations. Firstly, this is a cross-sectional, case-control study and, therefore, we cannot make conclusive causal deductions. Second, it would have been more interesting if we had assayed other immune and oxidative biomarkers of affective disorders, including pro-inflammatory cytokines, MDA, oxidized LDL, and PON1 activity.

### 4.4. Conclusions

The most important predictors of affective symptoms due to T2DM are in descending order of importance: copper, calcium, β-arrestin-1, LacCer, and FBXW7. T2DM and affective symptoms share common pathways, namely increased atherogenicity, IR, copper, and β-arrestin-1, and lowered calcium. Moreover, copper, calcium, β-arrestin-1, and LacCer (positive) and FBXW7 (inverse) may also induce affective symptoms by effects on IR and atherogenicity.

## Figures and Tables

**Figure 1 jpm-12-00023-f001:**
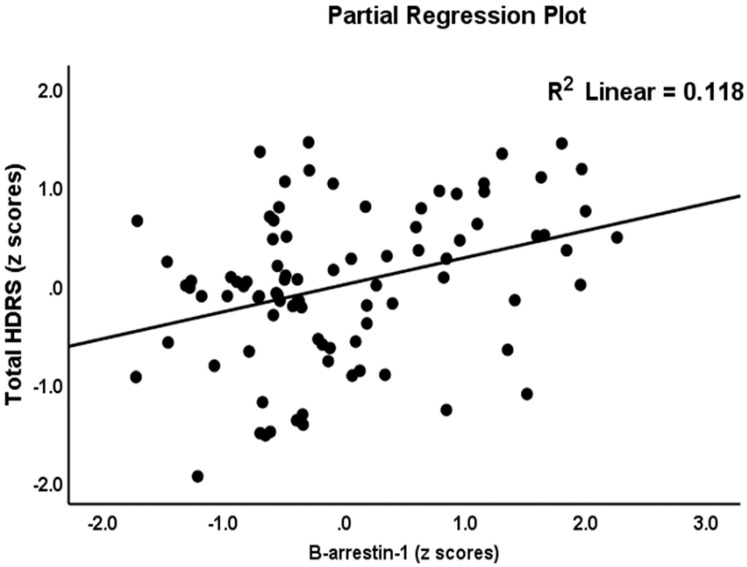
Partial regression plot with the total 17-item Hamilton Depression Rating Scale (HDRS) score as the dependent variable and β-arrestin-1 as explanatory variable.

**Figure 2 jpm-12-00023-f002:**
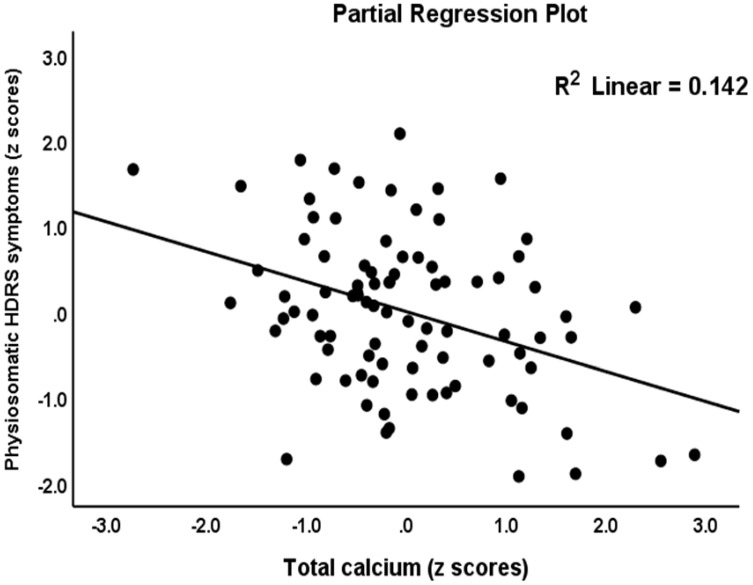
Partial regression plot with the physiosomatic component of the Hamilton Depression Score (HDRS) as dependent variable and total calcium as explanatory variable.

**Figure 3 jpm-12-00023-f003:**
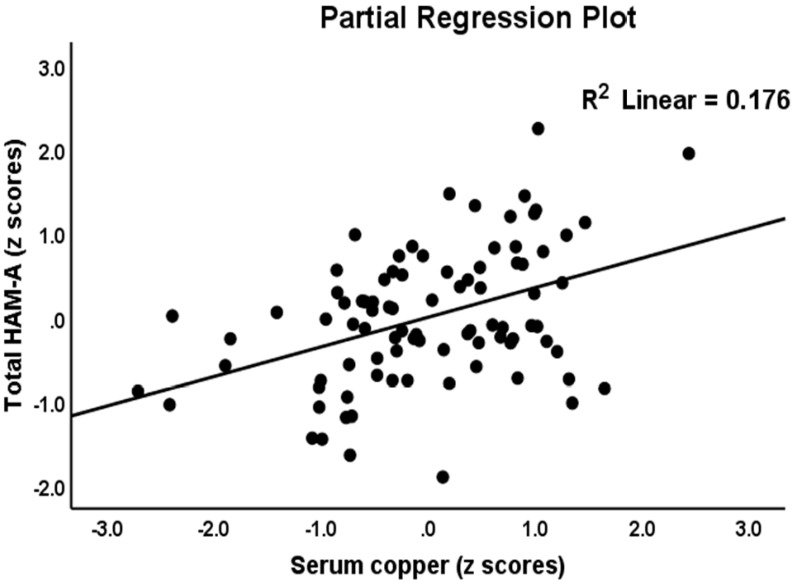
Partial regression plot with the Hamilton Anxiety Rating Scale (HAM-A) score as the dependent variable and serum copper as explanatory variable.

**Figure 4 jpm-12-00023-f004:**
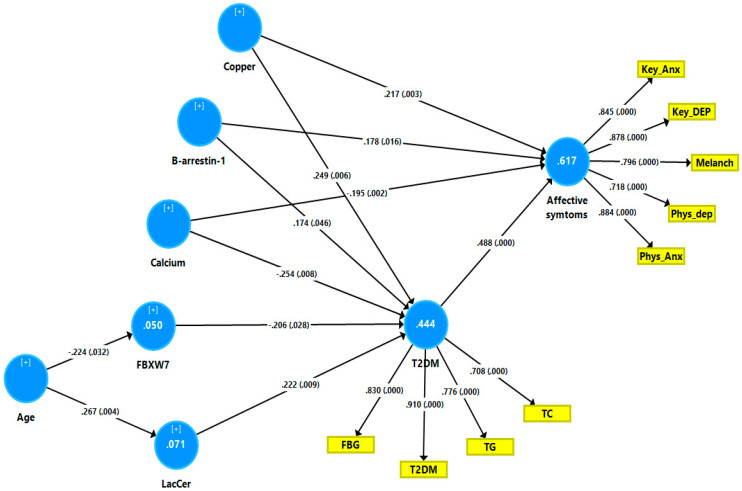
Results of partial Least Squares (PLS) path analysis. Shown are the significant path coefficients with exact *p*-values, and the explained variances (white figures in the circles). LacCer: lactosylceramide; FBG: fasting blood glucose; T2DM: type 2 diabetes mellitus; TG: triglycerides; TC: total cholesterol; Key_Dep: key depressive symptoms; Phys_Dep: physiosomatic symptoms of the HDRS (Hamilton Depression Rating Scale); Melanch: melancholic HDRS symptoms; Key_Anx: key anxiety symptoms of the Hamilton Anxiety Rating Scale (HAM-A); Phys_Anx: physiosomatic HAM-A symptoms.

**Table 1 jpm-12-00023-t001:** Socio-demographic and clinical data in patients with T2DM subdivided according to the insulin resistance index (IRI) values into normal IRI values, medium increased IRI, and very high IRI values.

Variables	Normal-IRI ^A^ (*n* = 30)	Increased-IRI ^B^ (*n* = 33)	Very High-IRI ^C^ (*n* = 25)	F/χ^2^	df	*p*
Age (years)	48.5 ± 5.5	48.6 ± 7.4	49.0 ± 5.2	0.06	2/85	0.944
Body mass index (kg/m^2^)	26.98 ± 3.29	27.50 ± 2.75	26.29 ± 4.57	0.83	2/85	0.439
Education (years)	10.0 ± 3.9	10.1 ± 4.6	9.3 ± 3.9	0.32	2/85	0.727
Single/Married	4/26	5/28	4/21	0.08	2	0.959
Rural/Urban	12/18	21/12	6/19	1.68	2	0.433
Employment Yes/No	17/13	17/16	12/13	0.42	2	0.809
Family history Yes/No	15/15 ^B,C^	12/21 ^A,C^	19/6 ^A,B^	9.05	2	0.011
Drug free/Diet only/drugs	16/6/8 ^C^	12/10/11 ^C^	0/9/16 ^A,B^	FEPT	-	0.001
FBG mM	6.79 ± 2.09 ^B,C^	9.53 ± 3.95 ^A,C^	13.29 ± 3.68 ^A,B^	25.75	2/85	<0.001
Insulin pM	42.80 ± 10.36 ^B,C^	53.45 ± 16.74 ^A,C^	62.14 ± 19.70 ^A,B^	10.27	2/85	<0.001
IRI (z score)	−1.073 ± 0.451 ^B,C^	0.002 ± 0.321 ^A,C^	1.196 ± 0.527 ^A,B^	188.25	2/85	<0.001
zβcell (z score)	0.057 ± 0.570	0.016 ± 1.113	−0.243 ± 1.143	0.76	2/85	0.472
Triglycerides mM	1.38 ± 0.47 ^B,C^	1.80 ± 0.57 ^A^	2.03 ± 0.59 ^A^	10.47	2/85	<0.001
Total cholesterol mmol/L	5.25 ± 0.89 ^B,C^	5.77 ± 0.97 ^A^	5.91 ± 0.80 ^A^	4.22	2/85	0.018
HDLc mmol/L	1.04 ± 0.15	1.05 ± 0.16	0.99 ± 0.16	0.85	2/85	0.432
LDLc mmol/L	3.58 ± 0.76	3.90 ± 0.84	3.99 ± 0.58	2.34	2/85	0.102
zCastelli (z scores)	−0.348 ± 0.983 ^C^	0.067 ± 1.110	0.458 ± 0.553 ^A^	5.09	2/85	0.008
zAIP (z scores)	−0.435 ± 0.800 ^B,C^	0.047 ± 1.040 ^A,C^	0.572 ± 0.885 ^A,B^	8.31	2/85	0.001

All results are shown as mean (SD); ^A,B,C^: pairwise comparisons between group means; F: results of analysis of variance; χ^2^: results of analysis of contingency analysis; FBG: fasting blood sugar; HDLc: high-density lipoprotein cholesterol; HDLc: low-density lipoprotein cholesterol; AIP: atherogenic index of plasma.

**Table 2 jpm-12-00023-t002:** Measurements of the Hamilton Depression Rating Scale (HDRS) and Hamilton Rating Scale for Anxiety (HAM-A) total and subdomain scores and biomarkers in patients with type 2 diabetes mellitus (T2DM), subdivided into those with normal insulin resistance index (IRI) values, medium increased IR, and very high IR values.

Variables	Normal-IRI (*n* = 28)	Increased-IRI (*n* = 33)	Very High-IRI (*n* = 25)	F/χ^2^	df	*p*
Total HDRS	8.33 ± 6.16 ^C^	10.30 ± 5.45 ^C^	13.20 ± 3.96 ^A,B^	5.68	2/85	0.005
Key HDRS	2.20 ± 1.83 ^C^	2.70 ± 1.61 ^C^	4.00 ± 1.50 ^A,B^	8.42	2/85	<0.001
Physiosomatic HDRS	2.47 ± 2.08 ^C^	3.24 ± 2.05	3.92 ± 1.98 ^A^	3.50	2/85	0.035
Melancholia HDRS	1.63 ± 1.65 ^C^	2.03 ± 1.69	2.64 ± 1.25 ^A^	2.84	2/85	0.064
Total HAM-A	9.17 ± 6.32 ^C^	10.67 ± 5.99 ^C^	14.92 ± 4.13 ^A,B^	7.47	2/85	0.001
Key HAM-A	2.77 ± 2.10 ^C^	3.15 ± 2.03 ^C^	4.28 ± 1.40 ^A,B^	4.59	2/85	0.013
Physiosomatic HAM-A	4.27 ± 3.14 ^C^	4.94 ± 3.08 ^C^	7.16 ± 2.81 ^A,B^	6.69	2/85	0.002
β-arrestin-1 ng/mL	12.81 ± 7.21	15.60 ± 7.86	17.39 ± 6.93	2.71	2/85	0.072
Serotonin ng/mL	143.1 ± 73.0	142.75 ± 81.1	184.4 ± 114.4	1.92	2/85	0.153
FBXW7 ng/mL	16.40 ± 0.62 ^C^	16.73 ± 8.40 ^C^	11.26 ± 6.71 ^A,B^	4.32	2/85	0.016
Lactosylceramide ng/mL	27.05 ± 11.35 ^C^	30.49 ± 15.01 C	38.80 ± 16.78 ^A,B^	4.69	2/85	0.012
Albumin g/L	45.47 ± 6.95	45.82 ± 5.47	46.68 ± 4.79	0.31	2/85	0.738
Total magnesium mM	0.736 ± 0.161 ^C^	0.682 ± 0.192	0.637 ± 0.124 ^A^	2.49	2/85	0.089
Total calcium mM	2.287 ± 0.156	2.259 ± 0.168	2.265 ± 0.179	0.24	2/85	0.791
Copper mg/L	0.975 ± 0.206	0.931 ± 0.242 ^C^	1.050 ± 0.113 ^B^	2.53	2/85	0.086
Zinc mg/L	0.716 ± 0.158	0.685 ± 0.200	0.638 ± 0.156	1.36	2/85	0.262

All results are shown as mean (SD); ^A,B,C^: pairwise comparisons between group means; HDRS: Hamilton Depression Rating Scale; HAM-A: Hamilton Rating Scale for Anxiety; F-box/WD repeat-containing protein 7 (FBXW7).

**Table 3 jpm-12-00023-t003:** Results of multiple regression with the Hamilton Depression Rating Scale (HDRS) scores as dependent variables and biomarkers as explanatory variables.

Dependent Variables	Explanatory Variables	Β	t	*p*	F Model	df	*p*	R^2^
#1a. Total HDRS-17	Model				19.03	4/83	<0.001	0.478
	Calcium	−0.358	−4.39	<0.001				
	Copper	0.273	3.25	0.002				
	β-arrestin-1	0.275	3.33	0.001				
	Lactosylceramide	0.187	2.19	0.031				
#1b. Total HDRS-17	Model				20.66	4/83	<0.001	0.499
	Calcium	−0.326	−3.99	<0.001				
	Copper	0.302	3.79	<0.001				
	β-arrestin-1	0.240	2.90	0.005				
	Castelli risk index 1	0.245	2.90	0.005				
#2a. Key_HDRS	Model				14.66	3/84	<0.001	0.344
	β-arrestin-1	0.327	3.60	0.001				
	Copper	0.283	3.12	0.002				
	Calcium	−0.271	−3.01	0.004				
#2b. Key_HDRS	Model				15.75	4/83	<0.001	0.432
	Castelli risk index 1	0.323	3.58	0.001				
	Copper	0.261	3.07	0.003				
	β-arrestin-1	0.243	2.76	0.007				
	Calcium	−0.194	−2.23	0.029				
#3a. Physiom_HDRS	Model				10.45	3/84	<0.001	0.272
	Calcium	−0.351	−3.73	<0.001				
	Copper	0.233	2.42	0.018				
	FBXW7	−0.216	−2.27	0.026				
#3b. Physiom_HDRS	Model				11.01	3/84	<0.001	0.282
	Calcium	−0.353	−3.78	<0.001				
	Copper	0.255	2.72	0.008				
	Insulin Resistance Index	0.236	2.54	0.013				
#4a. Melanch_HDRS	Model				10.10	3/84	<0.001	0.265
	Copper	0.285	2.98	0.004				
	β-arrestin-1	0.286	3.00	0.004				
	Zinc	−0.233	−2.49	0.015				
#4b. Melanch_HDRS	Model				11.55	4/83	<0.001	0.358
	Castelli risk index 1	0.334	3.59	0.001				
	Copper	0.228	2.49	0.015				
	Albumin	0.226	2.52	0.014				
	β-arrestin-1	0.195	2.08	0.041				

Key_Dep: key depressive symptoms; Phys_Dep: physiosomatic symptoms of the HDRS (Hamilton Depression Rating Scale); Melanch: melancholic HDRS symptoms; Key_Anx: key anxiety symptoms of the Hamilton Anxiety Rating Scale (HAM-A); Phys_Anx: physiosomatic HAM-A symptoms.

**Table 4 jpm-12-00023-t004:** Results of multiple regression analysis with the Hamilton Anxiety Rating Scale (HAM-A) scores as dependent variables and biomarkers as explanatory variables.

Dependent Variables	Explanatory Variables	β	t	*p*	F Model	df	*p*	R^2^
#1. HAM-A total score	Model				18.95	4/83	<0.001	0.477
	Copper	0.354	4.21	<0.001				
	β-arrestin-1	0.289	3.50	0.001				
	Calcium	−0.267	−3.27	0.002				
	Lactosylceramide	0.175	2.05	0.043				
#2. HAM-A total score	Model				23.34	4/83	<0.001	0.529
	Castelli risk index 1	0.306	3.72	<0.001				
	Copper	0.376	4.87	<0.001				
	β-arrestin-1	0.237	2.96	0.004				
	Calcium	−0.219	−2.76	0.007				
#3. Key_HAM-A	Model				14.33	5/82	<0.001	0.466
	Copper	0.366	4.32	<0.001				
	FBXW7	−0.307	−3.50	0.001				
	Calcium	−0.244	−2.96	0.004				
	β-arrestin-1	0.229	2.61	0.011				
	Age	−0.213	−2.51	0.014				
#4. Key_HAM-A	Model				13.21	6/81	<0.001	0.494
	Copper	0.360	4.34	<0.001				
	Castelli risk index 1	0.191	2.12	0.037				
	FBXW7	−0.252	−2.81	0.006				
	Calcium	−0.196	−2.33	0.022				
	Age	−0.201	−2.42	0.018				
	β-arrestin-1	0.195	2.23	0.029				
#5. Physiosom_HAM-A	Model				16.10	4/83	<0.001	0.437
	Copper	0.320	3.67	<0.001				
	Calcium	−0.281	−3.32	0.001				
	β-arrestin-1	0.256	2.98	0.004				
	Lactosylceramide	0.186	2.10	0.039				
#6. Physiosom_HAM-A	Model				18.70	4/83	<0.001	0.474
	Atherogenic index of plasma	0.284	3.25	0.002				
	Copper	0.327	3.96	<0.001				
	β-arrestin-1	0.231	2.77	0.007				
	Calcium	−0.228	−2.70	0.009				

Key_HAM-A: key anxiety symptoms of the HAM-A; Physiosom_HAM-A: physiosomatic HAM-A symptoms.

**Table 5 jpm-12-00023-t005:** Results of binary logistic regression analysis with T2DM as dependent variable and biomarkers as explanatory variables.

Dependent Variables	Explanatory Variables	B	SE	Wald	df	*p*	OR	95% CI
#1. T2DM Patients vs. Controls	Serotonin	1.224	0.386	10.05	1	0.002	3.40	1.60–7.25
	FBXW7	−1.107	0.414	7.15	1	0.008	0.33	0.15–0.74
	Lactosylceramide	1.929	0.568	11.52	1	0.001	6.88	2.26–20.96
	Calcium	−1.502	0.401	14.02	1	<0.001	0.22	0.10–0.49
	Copper	1.863	0.506	13.57	1	<0.001	6.44	2.39–17.37
#2. T2DM Patients vs. Controls	β-arrestin 1	1.014	0.317	10.26	1	0.001	2.76	1.48–5.13
#3. T2DM Patients vs. Controls	Magnesium	−1.082	0.302	12.80	1	<0.001	0.34	0.19–0.61

**Table 6 jpm-12-00023-t006:** Results of multiple regression analysis with insulin resistance, β-cell function, Castelli risk index 1, and atherogenic index of plasma (AIP) as dependent variables.

Dependent Variables	Explanatory Variables	β	t	*p*	F Model	df	*p*	R^2^
#1. Insulin resistance index	Model				7.26	2/85	0.001	0.146
	FBXW7	−0.239	−2.28	0.025				
	Lactosylceramide	0.237	2.26	0.026				
#2. β-cell function index	Model				13.410	5/82	<0.001	0.450
	B-arrestin	−0.284	−3.30	0.001				
	Calcium	0.324	3.79	<0.001				
	Copper	−0.277	−3.12	0.003				
	Albumin	0.251	2.93	0.004				
	Family history	−0.186	−2.00	0.049				
#3. Castelli Risk index 1	Model				11.70	3/84	<0.001	0.295
	Family history	0.325	3.38	0.001				
	FBXW7	−0.274	−2.93	0.004				
	Calcium	−0.207	−2.20	0.031				
#4. AIP	Model				8.94	3/84	<0.001	0.242
	Magnesium	−0.239	−2.38	0.020				
	Serotonin	0.282	2.97	0.004				
	Calcium	−0.247	−2.25	0.016				

## Data Availability

The dataset generated during and/or analyzed during the current study will be available from M.M. upon reasonable request and once the dataset has been fully exploited by the authors.

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
