# Peer review of "Intersections between Copper, β-Arrestin-1, Calcium, FBXW7, CD17, Insulin Resistance and Atherogenicity Mediate Depression and Anxiety Due to Type 2 Diabetes Mellitus: A Nomothetic Network Approach"

_jpm, 2022, doi:10.3390/jpm12010023_

Round 1

Reviewer 1 Report

Dear authors, thank you for the interesting study. 

There are some minor concerns.

I recommend the authors to follow the CONSORT guidelines for reporting clinical trials. There are several aspects of the checklist that were not followed in the present manuscript.

Description of the trial including allocation ratio. Please give more information on the design of the study. 

  1. How sample size was determined
  2. The randomization process is unclear and incomplete
  3. Why did the authors focus only on men?
  4. What kind of treatment did the DM patients receive? Could it have affected the results?

Author Response

REVIEWER 1

I recommend the authors to follow the CONSORT guidelines for reporting clinical trials. There are several aspects of the checklist that were not followed in the present manuscript.

Description of the trial including allocation ratio. Please give more information on the design of the study

@ANSWER: The present study is a case-control study and we added the study design in the Subjects & Methods section.

How sample size was determined

@ANSWER: We added a statement explaining the calculation of the sample size of the study.It reads:

A priori estimation of the required study sample showed that at least 70 individuals should be included to obtain a power of 0.8 with an effect size of 0.2, alpha level of 0.05, and 5 predictors in a linear multiple regression model.

The randomization process is unclear and incomplete

@ANSWER: The study is not a clinical trial. Neverthless, patients and controls were matched according to age, BMI and education. This is explained in the Subjects section:

The current case-control study recruited 58 T2DM male patients and 30 age, BMI and education matched healthy controls.

Why did the authors focus only on men?

@ANSWER: This is addressed in the text as:

We selected male subjects to exclude possible effects of the female hormonal status and the menstrual cycle.

What kind of treatment did the DM patients receive? Could it have affected the results?

@ANSWER: We address this remark in the text as:

In the patient group, 12 patients were drug free, 19 were on diabetes diet, and 27 were treated with glibenclamide 5 mg/day.

The drug/diet state of the patients was entered in all multiple linear regression analysys as  explanatory dummy variables. However, neither diabetes diet nor glibencamide showed a significant effect on the dependent variables.

REVIEWER 1

I recommend the authors to follow the CONSORT guidelines for reporting clinical trials. There are several aspects of the checklist that were not followed in the present manuscript.

Description of the trial including allocation ratio. Please give more information on the design of the study

@ANSWER: The present study is a case-control study and we added the study design in the Subjects & Methods section.

How sample size was determined

@ANSWER: We added a statement explaining the calculation of the sample size of the study.It reads:

A priori estimation of the required study sample showed that at least 70 individuals should be included to obtain a power of 0.8 with an effect size of 0.2, alpha level of 0.05, and 5 predictors in a linear multiple regression model.

The randomization process is unclear and incomplete

@ANSWER: The study is not a clinical trial. Neverthless, patients and controls were matched according to age, BMI and education. This is explained in the Subjects section:

The current case-control study recruited 58 T2DM male patients and 30 age, BMI and education matched healthy controls.

Why did the authors focus only on men?

@ANSWER: This is addressed in the text as:

We selected male subjects to exclude possible effects of the female hormonal status and the menstrual cycle.

What kind of treatment did the DM patients receive? Could it have affected the results?

@ANSWER: We address this remark in the text as:

In the patient group, 12 patients were drug free, 19 were on diabetes diet, and 27 were treated with glibenclamide 5 mg/day.

The drug/diet state of the patients was entered in all multiple linear regression analysys as  explanatory dummy variables. However, neither diabetes diet nor glibencamide showed a significant effect on the dependent variables.

REVIEWER 1

I recommend the authors to follow the CONSORT guidelines for reporting clinical trials. There are several aspects of the checklist that were not followed in the present manuscript.

Description of the trial including allocation ratio. Please give more information on the design of the study

@ANSWER: The present study is a case-control study and we added the study design in the Subjects & Methods section.

How sample size was determined

@ANSWER: We added a statement explaining the calculation of the sample size of the study.It reads:

A priori estimation of the required study sample showed that at least 70 individuals should be included to obtain a power of 0.8 with an effect size of 0.2, alpha level of 0.05, and 5 predictors in a linear multiple regression model.

The randomization process is unclear and incomplete

@ANSWER: The study is not a clinical trial. Neverthless, patients and controls were matched according to age, BMI and education. This is explained in the Subjects section:

The current case-control study recruited 58 T2DM male patients and 30 age, BMI and education matched healthy controls.

Why did the authors focus only on men?

@ANSWER: This is addressed in the text as:

We selected male subjects to exclude possible effects of the female hormonal status and the menstrual cycle.

What kind of treatment did the DM patients receive? Could it have affected the results?

@ANSWER: We address this remark in the text as:

In the patient group, 12 patients were drug free, 19 were on diabetes diet, and 27 were treated with glibenclamide 5 mg/day.

The drug/diet state of the patients was entered in all multiple linear regression analysys as  explanatory dummy variables. However, neither diabetes diet nor glibencamide showed a significant effect on the dependent variables.

REVIEWER 1

I recommend the authors to follow the CONSORT guidelines for reporting clinical trials. There are several aspects of the checklist that were not followed in the present manuscript.

Description of the trial including allocation ratio. Please give more information on the design of the study

@ANSWER: The present study is a case-control study and we added the study design in the Subjects & Methods section.

How sample size was determined

@ANSWER: We added a statement explaining the calculation of the sample size of the study.It reads:

A priori estimation of the required study sample showed that at least 70 individuals should be included to obtain a power of 0.8 with an effect size of 0.2, alpha level of 0.05, and 5 predictors in a linear multiple regression model.

The randomization process is unclear and incomplete

@ANSWER: The study is not a clinical trial. Neverthless, patients and controls were matched according to age, BMI and education. This is explained in the Subjects section:

The current case-control study recruited 58 T2DM male patients and 30 age, BMI and education matched healthy controls.

Why did the authors focus only on men?

@ANSWER: This is addressed in the text as:

We selected male subjects to exclude possible effects of the female hormonal status and the menstrual cycle.

What kind of treatment did the DM patients receive? Could it have affected the results?

@ANSWER: We address this remark in the text as:

In the patient group, 12 patients were drug free, 19 were on diabetes diet, and 27 were treated with glibenclamide 5 mg/day.

The drug/diet state of the patients was entered in all multiple linear regression analysys as  explanatory dummy variables. However, neither diabetes diet nor glibencamide showed a significant effect on the dependent variables.

REVIEWER 1

I recommend the authors to follow the CONSORT guidelines for reporting clinical trials. There are several aspects of the checklist that were not followed in the present manuscript.

Description of the trial including allocation ratio. Please give more information on the design of the study

@ANSWER: The present study is a case-control study and we added the study design in the Subjects & Methods section.

How sample size was determined

@ANSWER: We added a statement explaining the calculation of the sample size of the study.It reads:

A priori estimation of the required study sample showed that at least 70 individuals should be included to obtain a power of 0.8 with an effect size of 0.2, alpha level of 0.05, and 5 predictors in a linear multiple regression model.

The randomization process is unclear and incomplete

@ANSWER: The study is not a clinical trial. Neverthless, patients and controls were matched according to age, BMI and education. This is explained in the Subjects section:

The current case-control study recruited 58 T2DM male patients and 30 age, BMI and education matched healthy controls.

Why did the authors focus only on men?

@ANSWER: This is addressed in the text as:

We selected male subjects to exclude possible effects of the female hormonal status and the menstrual cycle.

What kind of treatment did the DM patients receive? Could it have affected the results?

@ANSWER: We address this remark in the text as:

In the patient group, 12 patients were drug free, 19 were on diabetes diet, and 27 were treated with glibenclamide 5 mg/day.

The drug/diet state of the patients was entered in all multiple linear regression analysys as  explanatory dummy variables. However, neither diabetes diet nor glibencamide showed a significant effect on the dependent variables.

REVIEWER 1

I recommend the authors to follow the CONSORT guidelines for reporting clinical trials. There are several aspects of the checklist that were not followed in the present manuscript.

Description of the trial including allocation ratio. Please give more information on the design of the study

@ANSWER: The present study is a case-control study and we added the study design in the Subjects & Methods section.

How sample size was determined

@ANSWER: We added a statement explaining the calculation of the sample size of the study.It reads:

A priori estimation of the required study sample showed that at least 70 individuals should be included to obtain a power of 0.8 with an effect size of 0.2, alpha level of 0.05, and 5 predictors in a linear multiple regression model.

The randomization process is unclear and incomplete

@ANSWER: The study is not a clinical trial. Neverthless, patients and controls were matched according to age, BMI and education. This is explained in the Subjects section:

The current case-control study recruited 58 T2DM male patients and 30 age, BMI and education matched healthy controls.

Why did the authors focus only on men?

@ANSWER: This is addressed in the text as:

We selected male subjects to exclude possible effects of the female hormonal status and the menstrual cycle.

What kind of treatment did the DM patients receive? Could it have affected the results?

@ANSWER: We address this remark in the text as:

In the patient group, 12 patients were drug free, 19 were on diabetes diet, and 27 were treated with glibenclamide 5 mg/day.

The drug/diet state of the patients was entered in all multiple linear regression analysys as  explanatory dummy variables. However, neither diabetes diet nor glibencamide showed a significant effect on the dependent variables.

REVIEWER 1

I recommend the authors to follow the CONSORT guidelines for reporting clinical trials. There are several aspects of the checklist that were not followed in the present manuscript.

Description of the trial including allocation ratio. Please give more information on the design of the study

@ANSWER: The present study is a case-control study and we added the study design in the Subjects & Methods section.

How sample size was determined

@ANSWER: We added a statement explaining the calculation of the sample size of the study.It reads:

A priori estimation of the required study sample showed that at least 70 individuals should be included to obtain a power of 0.8 with an effect size of 0.2, alpha level of 0.05, and 5 predictors in a linear multiple regression model.

The randomization process is unclear and incomplete

@ANSWER: The study is not a clinical trial. Neverthless, patients and controls were matched according to age, BMI and education. This is explained in the Subjects section:

The current case-control study recruited 58 T2DM male patients and 30 age, BMI and education matched healthy controls.

Why did the authors focus only on men?

@ANSWER: This is addressed in the text as:

We selected male subjects to exclude possible effects of the female hormonal status and the menstrual cycle.

What kind of treatment did the DM patients receive? Could it have affected the results?

@ANSWER: We address this remark in the text as:

In the patient group, 12 patients were drug free, 19 were on diabetes diet, and 27 were treated with glibenclamide 5 mg/day.

The drug/diet state of the patients was entered in all multiple linear regression analysys as  explanatory dummy variables. However, neither diabetes diet nor glibencamide showed a significant effect on the dependent variables.

Reviewer 2 Report

The article by Al-Hakeim et al., "Intersections between copper, β-arrestin-1, calcium, FBXW7, CD17, insulin resistance and atherogenicity mediate depression and anxiety due to type 2 diabetes mellitus: a nomothetic network approach" delineated common markers between T2DM and affective disorders. The article is novel and well written. 

But I have some concerns; 

1. How would the authors justify the significance of this study with such a sample size, 58 T2DM vs 30 controls

2. Why were only male patients included in this study? 

Author Response

The article by Al-Hakeim et al., "Intersections between copper, β-arrestin-1, calcium, FBXW7, CD17, insulin resistance and atherogenicity mediate depression and anxiety due to type 2 diabetes mellitus: a nomothetic network approach" delineated common markers between T2DM and affective disorders. The article is novel and well written. 

But I have some concerns; 

  1. How would the authors justify the significance of this study with such a sample size, 58 T2DM vs 30 controls

@ANSWER: The number of subjects was based on a priori estimation of the sample size to obtain a power of 0.8. Nevertheless, increasing the study sample would improve the estimated parameters. Discussed in the text as:

A priori estimation of the required study sample showed that at least 70 individuals should be included to obtain a power of 0.8 with an effect size of 0.2, alpha level of 0.05, and 5 predictors in a linear multiple regression model. The same power analysis can be applied when examining PLS path analysis indicating that the power of this PLS analysis was > 0.8 [72].

  1. Why were only male patients included in this study? 

@ANSWER: addressed in the text as:

We selected male subjects to exclude possible effects of the female hormonal status and the menstrual cycle.